# Hexokinase-II Inhibition Synergistically Augments the Anti-tumor Efficacy of Sorafenib in Hepatocellular Carcinoma

**DOI:** 10.3390/ijms20061292

**Published:** 2019-03-14

**Authors:** Jeong-Ju Yoo, Su Jong Yu, Juri Na, Kyungmin Kim, Young Youn Cho, Yun Bin Lee, Eun Ju Cho, Jeong-Hoon Lee, Yoon Jun Kim, Hyewon Youn, Jung-Hwan Yoon

**Affiliations:** 1Department of Internal Medicine, Soonchunhyang University Bucheon Hospital, Gyeonggi-do 14584, Korea; puby17@naver.com; 2Department of Internal Medicine and Liver Research Institute, Seoul National University College of Medicine, Seoul National University Hospital, Seoul 03080, Korea; ydoctor2@hanmail.net (S.J.Y.); yunbin@hanmail.net (Y.B.L.); creatioex@gmail.com (E.J.C.); pindra@empal.com (J.-H.L.); yoonjun@snu.ac.kr (Y.J.K.); 3Department of Nuclear Medicine, Cancer Research Institute, Seoul National University College of Medicine, Seoul National University Hospital, Seoul 03080, Korea; najul@snu.ac.kr (J.N.); koyo117@snu.ac.kr (K.K.); 4Department of Internal Medicine, Chung-Ang University Hospital, Seoul 03080, Korea; yycho@caumc.or.kr

**Keywords:** hepatocellular carcinoma, sorafenib, hexokinase inhibitor, bromopyruvate

## Abstract

This study aimed to examine whether inhibition of hexokinase (HK)-II activity enhances the efficacy of sorafenib in in-vivo models of hepatocellular carcinoma (HCC), and to evaluate the prognostic implication of HK-II expression in patients with HCC. We used 3-bromopyruvate (3-BP), a HK-II inhibitor to target HK-II. The human HCC cell line was tested as both subcutaneous and orthotopic tumor xenograft models in BALB/c nu/nu mice. The prognostic role of HK-II was evaluated in data from HCC patients in The Cancer Genome Atlas (TCGA) database and validated in patients treated with sorafenib. Quantitative real-time PCR, western blot analysis, and immunohistochemical staining revealed that HK-II expression is upregulated in the presence of sorafenib. Further analysis of the endoplasmic reticulum-stress network model in two different murine HCC models showed that the introduction of additional stress by 3-BP treatment synergistically increased the in vivo/vitro efficacy of sorafenib. We found that HCC patients with increased HK-II expression in the TCGA database showed poor overall survival, and also confirmed similar results for TCGA database HCC patients who had undergone sorafenib treatment. These results suggest that HK-II is a promising therapeutic target to enhance the efficacy of sorafenib and that HK-II expression might be a prognostic factor in HCC.

## 1. Introduction

Hepatocellular carcinoma (HCC) is one of the leading worldwide causes of cancer-related deaths and is the second most common cancer in men worldwide [1]. Because many patients are still diagnosed at an advanced stage, and there is no effective systemic therapy, the median survival of untreated patients remains poor [2]. Sorafenib is the only approved systemic therapy for patients with advanced HCC [3]; however, sorafenib monotherapy confers a modest gain in survival compared to placebo [4,5]. Moreover, the prognostic factors of patients undergoing sorafenib treatment and the mechanisms that mediate sorafenib resistance have not been fully revealed [6].

Several studies have demonstrated that a variety of mechanisms are involved in sorafenib resistance, which includes increased MAPK14 activity [7], activation of PI3K/AKT signaling [8], overexpression of CD44 [9], and enhanced glycolysis [10,11,12]. Studies on cellular metabolism and glycolysis have provided intriguing information for sorafenib therapy, which has been shown to lead to the inhibition of oxidative phosphorylation and the enhancement of glycolysis in a subset of HCC cell lines [10,13]. In the glycolytic pathway, hexokinase (HK)-II catalyzes the first irreversible and rate-limiting step [14], but data on the prognostic role of HK-II activity after sorafenib for patients with HCC are scarce.

We hypothesized that upregulated HK-II expression not only leads to the reduced efficacy of sorafenib but is also a predictor of tumor resistance to sorafenib. In order to investigate whether or not the upregulation of HK-II expression affects the efficacy of sorafenib, we analyzed the activity of sorafenib in HK-II overexpressing human HCC cell lines in in-vivo mouse models. In addition, we retrospectively evaluated whether or not HK-II expression in the tumor tissues of sorafenib-treated patients with HCC was a prognostic indicator.

In this study, we show that upregulated HK-II reduced sorafenib-induced apoptotic cell death and that 3-bromopyruvate (3-BP), an HK-II inhibitor [15], attenuated the reduction. We also produce evidence from murine HCC models on the beneficial effect of therapy that suppresses the effect of upregulated HK-II on sorafenib treatment in various murine HCC models. Moreover, we show that upregulated HK-II predicts poor outcomes in patients with HCC who underwent sorafenib treatment. These findings have important implications when planning sorafenib-based therapy in patients with HCC.

## 2. Results

### 2.1. Sorafenib Treatment Leads to Upregulation of Hexokinase-II Expression in Tumor Cell Cultures

We first investigated whether 3-BP combined with sorafenib inhibited proliferation of HCC cells in vitro. For this purpose, human HCC cell lines, SNU-761 (Appendix A) and Huh-7 (Appendix A) were cultured in normoxic and hypoxic state. Sorafenib inhibited cellular growth in a dose-dependent manner, and this effect of growth inhibition was amplified when combined with 3-BP, both in normoxic and hypoxic conditions. In short, 3-BP in combination with sorafenib strongly inhibited primary tumor growth.

Since the anti-angiogenic effect of sorafenib might induce tumor cells to rely increasingly on glycolysis instead of oxidative phosphorylation, we evaluated culture supernatant lactate levels as a surrogate marker for glycolysis. As shown in Appendix A, the cells receiving sorafenib significantly increased lactate production, which indicated increased glycolysis compared to the control cells under normoxic culture conditions (*p* = 0.032). We assessed whether the increased glycolysis after sorafenib was HK-II dependent and found that increased glycolysis after sorafenib was reversed by 3-BP, an HK-II inhibitor (*p* = 0.024) (Appendix A). Taken together, the results showed that sorafenib resulted in increased glycolysis in an HK-II dependent manner, and HK-II can be targeted by 3-BP.

Next, we tested the effect of sorafenib on HK-II expression in a preclinical subcutaneous HCC murine model of SNU-761 tumors. Sorafenib administration was initiated two weeks after injection of SNU-761 cells (when tumors were palpable (50 mm^3^)). As shown in Figure 1A, sorafenib caused marginal inhibition of subcutaneous tumors four weeks after the injection of tumor cells (*p* = 0.048). Immunohistochemical analysis showed that HK-II expression was significantly increased in tumors after sorafenib treatment (*p* < 0.0001, Figure 1B).

### 2.2. Hypoxia Inhibits the Efficacy of Sorafenib Treatment in HCC

The upregulation of HK-II expression associated with sorafenib treatment prompted us to test the relationship between hypoxia and sorafenib, an anti-angiogenic agent, in more detail. Since hypoxia is reported to induce HK-II expression [16], we investigated whether hypoxia inhibited the effect of sorafenib on the proliferation of HCC in vitro. The human HCC cell line SNU-761 was cultured under normoxic and hypoxic conditions in the presence or absence of sorafenib [17]. Under the normoxic condition, sorafenib effectively inhibited cellular proliferation (*p* = 0.0074, Figure 2A). By contrast, the hypoxic condition led to significant cellular proliferation compared to the normoxic condition, regardless of the presence of sorafenib (2 μM) (Figure 2B).

### 2.3. Hexokinase-II Inhibition by 3-BP Rescues Sorafenib Efficacy from Reduced ER Stress

We then studied the mechanism by which HK-II inhibits the effect of sorafenib. Hexokinase-II, which catalyzes the first step of the major survival pathway induced by hypoxia, was assessed with the use of 3-BP. Because sorafenib exerts its effect on ER stress due to its anti-angiogenic activity, and hypoxia induces HK-II, we postulated that the simultaneous administration of 3-BP and sorafenib would enhance the level of sorafenib-induced apoptosis in HCC cells. When 3-BP was treated with sorafenib, caspase-9 and -3 were more prominent than in the cells treated with just sorafenib alone, indicating that the activation of mitochondrial apoptotic signals was augmented by 3-BP (Figure 3A).

We then investigated kinase signals that regulate apoptosis and found that pro-apoptotic JNK was more highly activated in cells treated with sorafenib + 3-BP than in cells treated with sorafenib alone (Figure 3A). Because JNK activation might depend on ER stress, we investigated the activation of ER stress in cells treated with sorafenib + 3-BP. Indeed, eIF2α phosphorylation, which reflects activation of ER stress, was prominent in cells co-treated with sorafenib + 3-BP.

Other ER stress-mediated apoptotic markers, namely, *GADD34* and *GADD153* [18,19], were also markedly increased in cells treated with sorafenib + 3-BP, compared to sorafenib alone (Figure 3B). Taken together, these findings suggest that 3-BP might augment ER stress-dependent JNK activation in sorafenib-treated HCC cells, which thus leads to apoptosis of these cells.

### 2.4. 3-BP Augments the Antitumor Efficacy of Sorafenib In Vivo

We then tested whether or not 3-BP treatment enhances the activity of sorafenib against HCC in BALB/c-nu/nu mice bearing subcutaneous SNU-761 tumors. Compared to the control animals, therapy consisting of sorafenib alone led to only mild suppression of tumor growth (Figure 4). By contrast, the combination therapy of 3-BP and sorafenib led to the obvious inhibition of tumor growth (Figure 4). After 28 days of treatment, the volumes of the tumors in the groups treated with sorafenib, 3-BP, or combined sorafenib + 3-BP were reduced to 66.5%, 44.6%, and 39.4%, respectively, of the mean tumor size of the untreated control group (*p* = 0.021, Figure 4). The synergistic effect was evaluated by NONMEM, based on the assumption that growth of the tumors was exponential. The growth rate constants were significantly larger for the tumors of the mice treated by sorafenib + 3-BP than for the untreated control mice, and mice treated by sorafenib, demonstrating that 3-BP and sorafenib acted synergistically to inhibit tumor growth (Appendix A).

Next, we established an orthotopic HCC model by direct subcapsular injection of SNU-761 cells into the livers of BALB/c-nu/nu livers as a model of HCC patients to study the effect of combined therapy. To visualize tumor growth in the orthotopic HCC mouse model by in vivo imaging, an SNU-761-luc cell line was established. The strength of the luciferase signal corresponded to the number of viable cancer cells (*R*^2^ = 0.9978). Consistent with our observations of the subcutaneous tumor model, imaging analysis showed that sorafenib alone or 3-BP alone inhibited progression of the liver tumors. However, the combination treatment of 3-BP + sorafenib showed the strongest anti-tumor effect (Appendix A). After 28 days of treatment, the bioluminescent signals of the mice receiving sorafenib, 3-BP, or sorafenib + 3-BP were reduced to 85.1%, 36.6% and 19.9%, respectively, of the signals of the controls (*p* < 0.05, Figure 5A). Analysis of BLI images was carried out by Living Image 2.50 software (PerkinElmer, Waltham, MA, USA). The bioluminescent signals from tumors were also compared after autopsy (Figure 5B). The upper and middle panels show mouse organs and the sizes of removed livers after abdominal cuts. The bottom panel shows BLI signals that indicate the presence of injected cells and cellular proliferation. Similar to the results from the subcutaneous model, the combination of 3-BP and sorafenib improved efficacy of sorafenib-based anticancer therapy on HCC. Notably, 3-BP enhanced the therapeutic effect of sorafenib by approximately 40% (*p* < 0.05, Figure 5C). Together, the results obtained from two different HCC murine models, namely, subcutaneous and orthotopic tumors, support the concept that the targeting of HK-II should enhance the efficacy of sorafenib therapy against HCC.

We then investigated whether or not 3-BP combined with sorafenib enhanced apoptosis in mouse HCC tumors. We used the TUNEL staining method to quantify apoptosis. The percentage of TUNEL-stained cells was significantly higher in the tumors of mice receiving sorafenib + 3-BP than in the tumors of mice receiving sorafenib alone (*p* < 0.05, Figure 5D). We then explored immunohistochemical staining with anti-JNK (*p* < 0.05, Figure 5E) and anti-caspase-3 (*p* < 0.05, Figure 5F). The intensity of caspase-3 and JNK expression was significantly higher in the tumors of mice receiving sorafenib + 3-BP than in the tumors of mice receiving sorafenib alone. These findings suggest that 3-BP augmented ER-stress-dependent JNK activation in sorafenib-treated HCC-bearing mice. The microscopy image of each immunohistochemistry is shown in Appendix A.

### 2.5. Prognostic Role of Hexokinase-II Expression on Survival of Patients with HCC Patient Survival in the TCGA Database

A total of 224 eligible HCC patients were included in the study. Appendix A summarizes the clinicopathological characteristics of these patients. The median age was 61 years (range 17–85 years). There were 162 (72.3%) patients with N0, 3 (1.3%) patients with N1, and 59 (26.3%) patients with unknown stage disease. Most of the patients (172, 76.8%) did not have distant metastases, 3 (1.3%) had distant metastases, and the other 49 (21.9%) patients had unknown metastatic status. We stratified the patients into two groups according to the degree of HK-II expression, using a cutoff point of 0.2054, which provided the maximum sum of specificity and sensitivity for predicting overall survival (OS) (sensitivity, 57.14%; specificity, 63.39%; area under the curve (AUC), 0.548; 95% CI, 0.457–0.637; *p* = 0.58). At the end of the last follow-up, 68 patients had died of the disease, and the proportion of patients who died of the disease showed high HK-II expression levels in their tumors than those were alive. The median OS rates were 56.5 months (interquartile range (IQR) = 12.2–41.4) and 84.7 months (IQR = 13.8–45.6), for the HK-II-high and HK-II-low patients, respectively (hazard ratio [HR], 1.71; 95% confidence interval [CI], 1.02 to 2.86; *p* = 0.043) (Appendix A).

### 2.6. Elevated High Hexokinase-II Expression Predicts a Poor Clinical Outcome in Patients with HCC Who Have Undergone Sorafenib Treatment

Based on our results, we postulated that HK-II expression and activity are correlated with sorafenib resistance.

To examine the relationship between HK-II expression and sorafenib sensitivity in patients with HCC, we analyzed the degree of HK-II immunopositivity in HCC patients who had been treated with sorafenib (Appendix A). Immunohistochemical analysis of HK-II protein expression in archived tumor specimens from our HCC patient cohort (*n* = 94) demonstrated that HK-II expression was increased in the tumor tissue of 72 cases (76.6%), whereas 22 cases (23.4%) showed decreased HK-II expression compared to adjacent nonmalignant liver tissue (Appendix A).

The prognostic impact of HK-II expression in HCC tissues of patients treated with sorafenib was examined. As shown in Figure 6A, the Kaplan–Meier curve shows a significant prolongation of the time to progression (TTP) in patients with tumors showing low HK-II expression, compared with the TTP of the patients with high HK-II expression (log-rank, *p* = 0.048). High HK-II expression was independently associated with shorter TTP (adjusted HR (aHR) 1.909; 95% CI, 1.086–3.342; *p* = 0.026) after adjustment for age (Appendix A). Kaplan–Meier analysis also found a significant prolongation of OS in patients with tumors showing low HK-II expression compared with the OS of patients with high HK-II expression (log-rank, *p* < 0.001; Figure 6B). High HK-II expression was independently associated with poor OS (aHR 1.882; 95% CI, 1.171–3.190; *p* = 0.024; R_squared by Cox & Snell 0.995) after adjustment for Child–Pugh score, tumor number, and lymph node involvement (Appendix A).

## 3. Discussion

In this study, we showed that upregulated HK-II expression inhibited sorafenib-induced apoptotic cell death and that this finding could be reversed by 3-BP [15], an HK-II inhibitor. Using various murine HCC models, we also provided evidence of the beneficial effect on sorafenib treatment of added therapy that suppressed HK-II expression. Moreover, we showed that upregulated HK-II expression predicted poor outcomes in patients with HCC who had undergone sorafenib treatment. These findings have important implications for considering sorafenib-based therapy for patients with HCC.

The main finding of this study is the augmented in vitro and in vivo (HCC animal model) anti-tumor effect provided by 3-BP + sorafenib. This synergistic effect was attributed to the following two mechanisms: (i) the inhibition of aerobic glycolysis, which is enhanced after sorafenib exposure, resulting in energy depletion and apoptosis; and (ii) the promotion of ER stress due to sorafenib, also resulting in augmented apoptosis. Furthermore, the upregulation of HK-II expression in HCC tissue predicted poor survival of patients from The Cancer Genome Atlas (TCGA) database. We also found that HK-II expression in the tumors of HCC patients predicted resistance to sorafenib.

Although sorafenib is the only approved systemic chemotherapy for HCC, it has only shown modest results for OS [4]. Due to genetic diversity, HCC often shows primary resistance to sorafenib [20]. Hepatocellular carcinoma also frequently shows secondary resistance associated with sorafenib administration, because of the activation of compensatory pathways, such as the PI3K/Akt and JAK-STAT pathways, EMT, and tumor hypoxia [21]. We previously reported that hypoxia induced HK-II expression in HCC cells [14]. In contrast to normal cells, which mostly rely on mitochondrial oxidative phosphorylation to generate energy, most cancer cells rely on aerobic glycolysis, which is known as the ‘Warburg Effect’ [22]. Indeed, many HCC cell lines are known to rely mainly on aerobic glycolysis for generation of ATP, in an HK-II-dependent manner [23].

Inhibitors of aerobic glycolysis via the Warburg Effect have been reported to be effective, and some of the inhibitors are under clinical study [24]. When aerobic glycolysis was blocked, apoptosis was induced mainly by the decreased levels of ATP in cancer cells [10]. Our study also showed that the HK-II inhibitor, 3-BP, which inhibits glycolysis, is effective, which was demonstrated by a lactate assay. Apoptosis was likely induced by the markedly increased depletion of ATP that was associated with the use of the combination therapy of sorafenib + 3-BP. Indeed, sorafenib, unlike other multikinase inhibitors, has been reported to decrease ATP levels [10]. However, the effect of 3-BP might also be explained by its interactions with the binding sites of other enzymes with anti-kinase activity, but these phenomena still need further research [10].

Increased ER stress is one of the recently observed mechanisms of the antitumor actions of sorafenib. Previous studies have suggested that sorafenib increases ER stress, which leads to caspase-3 induced cell death [19]. Hexokinase-II inhibitors also show activity associated with ER stress. We have previously verified that 3-BP, a type of HK-II inhibitor, increases ER stress, which has also been documented by other investigators [25,26,27]. The agent 3-BP was first studied as an anti-cancer agent more than a decade ago at Johns Hopkins [28]. We previously reported that 3-BP effectively inhibited the in vitro and in vivo growth of HCC by causing the dissociation of HK-II from the permeability transition pore complex (PTPC), which activates mitochondrial apoptotic signals [29]. Unlike normal liver tissue, human HCC cells tend to overexpress HK-II, and HK-II overexpression becomes more prominent in a hypoxic environment (e.g., advanced infiltrative hypovascular HCC or after sorafenib administration) [14]. Therefore, 3-BP treatment might prove to be a very selective anticancer therapy, because HK-II is maximally expressed in HCC cells and minimally expressed in normal liver tissue [29]. The efficacy of 3-BP as a therapeutic agent was proven by several animal studies, including our previous investigations [29,30,31].

In our patient cohort, HK-II expression was significantly correlated with OS and TTP in patients who had undergone sorafenib treatment. However, we could not find an association between HK-II expression and different stage or tumor grade. Additional studies of other patient cohorts are needed to verify our findings. The relationship between other known resistance factors such as HIF-1α and VEGFR, and OS and TTP should also be investigated [32,33]. In addition, a patient-derived xenograft model might be useful to confirm the efficacy of sorafenib combined with an HK-II inhibitor as treatment for HCC.

In conclusion, HK-II is a useful therapeutic target that can lead to enhancement of the efficacy of sorafenib treatment. HK-II expression might also be useful as a marker that predicts sensitivity to sorafenib and clinical outcomes.

## 4. Materials and Methods

### 4.1. Cell Lines

The SNU-761 human HCC cell line [34] was maintained in RPMI-1640 medium (WelGene Inc., Seoul, Korea) containing 10% heat-inactivated fetal bovine serum and 1% penicillin/streptomycin at 37 °C in a humidified incubator with 5% CO_2_. The culture medium was replaced every three days.

A lentivirus was used for the stable transduction of the firefly luciferase-expressing SNU-761 cells. Stock virus was generated in the 293FT cell line, which was co-transfected via Lipofectamine^®^ 2000 (Invitrogen, Carlsbad, CA, USA) with the pMSCV/luciferase vector (5 μg), a gag-pol vector (5 μg), and an envelope vector (5 μg). Supernatants were collected at 48 h posttransfection and filtered with 0.45-μm polyvinylidene difluoride (PVDF) filters (Millipore, Billerica, MA, USA). Virus supernatants were tittered and stored at −80 °C until required. For viral transduction of cancer cells, virus containing medium was added to target cells and polybrene (hexadimethrine bromide, 8 μg/mL) or protamine sulfate was also added to the target cells. Luciferase-expressing cancer cells were pooled after puromycin selection.

### 4.2. Chemicals and Reagents

The agent 3-BP was obtained from Sigma-Aldrich, Inc. (St. Louis, MO, USA). Sorafenib was purchased from LC laboratories (Woburn, MA, USA).

### 4.3. Lactate Assay

Lactate production was used as a surrogate marker of glycolysis [11]. The SNU-761 cells were exposed to vehicle alone (control group), sorafenib alone (8 μM), 3-BP alone (75 μM) or both sorafenib (8 μM) + 3-BP (75 μM) for 3 h. The levels of extracellular lactate, the end product of glycolysis, were measured by a lactate colorimetric/fluorometric assay (Kit #K607; BioVision, CA, USA).

### 4.4. Reverse Transcription PCR and Quantitative Real-time PCR

The expression of genes induced by ER stress, GADD34 and GADD153, was monitored [18,19]. Total RNA was extracted with the RNeasy Plus Micro Kit (Qiagen, Venlo, The Netherlands). The cDNA was synthesized with the use of the 1st Strand cDNA Synthesis Kit (Roche Applied Science, Penzberg, Upper Bavaria, Germany). The Applied Biosystems Power SYBR Green Master Mix System (Life Technologies, Carlsbad, CA, USA) was used for quantitative PCR. Each sample was analyzed in duplicate. Primer sequences are shown in Appendix A. The identities of all PCR products were verified by sequencing.

### 4.5. Immunoblot Assay

Immunoblot assays were performed in the same manner as described in previous studies [25,35]. Briefly, cells were lysed for 20 min on ice with lysis buffer (50 mM Tris-HCl, pH 7.4; 1% Nonidet P-40; 0.25% sodium deoxycholate; 150 mM NaCl; 1 mM EDTA; 1 mM phenylmethylsulfonyl fluoride; 1 µg/L each of aprotinin, leupetin, pepstatin; 1 mM Na_3_VO_4_; 1 mM NaF) and centrifuged at 14,000× *g* for 10 min at 4 °C. Samples were resolved by sodium dodecyl sulfate polyacrylamide gel electrophoresis, transferred to nitrocellulose membranes, exposed to appropriate primary antibodies, and incubated with peroxidase-conjugated secondary antibodies (Biosource International, Camarillo, CA, USA). Bound antibodies were visualized via the use of a chemiluminescent substrate (ECL; Amersham, Arlington Heights, IL, USA) and exposed to Kodak X-OMAT film. The primary antibodies used in this study were as follows: mouse anti-phospho-eukaryotic initiation factor 2α (eIF2α) and anti-actin were obtained from Santa Cruz Biotechnology, Inc. (Santa Cruz, CA, USA); rabbit anti-caspase-9, rabbit anti-caspase-7 and mouse anti-phospho-JNK were obtained from Cell Signaling Technology, Inc. (Danvers, MA, USA).

### 4.6. Animal Studies

Male BALB/c nu/nu mice were purchased from Orient-Bio (Seongnam, Korea). All animal experiments were approved by the Institutional Animal Care and Use Committee at Seoul National University Hospital (SNUH-IACUC). To establish subcutaneous tumors, 1 × 10^6^ cells of SNU-761 in 40 μL of Matrigel were injected into the left flank of 7-week-old mice. Mice with 150- to 200-mm^3^ tumors were randomly assigned to 4 groups (8 mice per group). Tumor volume was measured by calipers every week and calculated by the following formula V = (length [mm] × width [mm]^2^) × 0.5 [mm]^3^. Orthotropic tumors were established by injecting 2.5 × 10^6^ cells ofluciferase-expressingSNU-761 in 40 μL of PBS under the subcapsular left liver lobe of anesthetized mice after subcostal mini laparotomy. Six days after the procedure, the mice were randomized into the following 4 groups (control, sorafenib, 3-BP, and sorafenib + 3-BP) based on the adequate luciferase signals as measured byIVIS 100 (total flux > 1 × 10^6^ photon/s/cm^2^/sr). An IVIS100 imaging system (Caliper Life Sciences, Hopkinton, MA, USA) was used for the acquisition of bioluminescent images. D-luciferin potassium salt (0.3 mg/mL in saline before use) was used as a substrate for luciferase, and 100 μL of the D-luciferin solution was injected intraperitoneally into the mice. Bioluminescent images were serially acquired until maximum signals were obtained. To quantify emitted light, regions of interest (ROI) were drawn over the regions of the tumors. Total photon flux was expressed as photons per cm^2^ per second per steradian (p/cm^2^/s/sr). Drugs (1.25 mg/kg/day for sorafenib, 1 mg/kg/day for 3-BP) were injected intraperitoneally every 5 days for 4 weeks [25,36]. Bioluminescent images were acquired and the weights of mice were also obtained, at baseline, 7, 14, 21, and 28 days after the tumor cells were injected. Specimens of tumor tissue were fixed in 10% formaldehyde and subjected to immunohistochemical staining and TUNEL assay. Tumor growth kinetics, quantitation of apoptosis, and immunohistochemical analysis of the specimens obtained from the mice studies are described in the supporting information.

### 4.7. Human Study

#### Role of HK-II Expression in the Survival of HCC Patients in the TCGA Database

Hexokinase-II mRNA expression levels were retrieved from the TCGA RNA sequence database (http://genome-cancer.ucsc.edu/). The study included patients who were histopathologically diagnosed with HCC, had not undergone pretreatment, and had complete data on OS [37].

The eligibility criteria, treatment regimens, and assessment of the response to sorafenib by patients with HCC are reported in the supporting information.

### 4.8. Statistical Analysis

All in-vitro experimental data were obtained from at least 3 independent cell line experiments from a minimum of 3 separate isolations and are expressed as means ± standard deviation (SD). The Mann–Whitney U was used to determine differences between groups. For the retrospective patient study, the TTP was calculated from the first day of sorafenib administration to the date of progression. Overall survival was calculated from the date of first administration of sorafenib to the date of death or last contact. Conventional clinical factors at the time of entry into the study and immunopositivity for HK-II were analyzed to identify variables that affected survival as determined by the Kaplan–Meier method and compared by the log-rank test. Stepwise multivariate analysis was performed using the Cox proportional hazards model to identify independent variables that affected survival. Factors found to be significantly related to outcome by univariate analysis were included in the multivariate analysis. For statistical analysis, SPSS version 21.0 for Windows (SPSS, Inc., Chicago, IL, USA) was used. Where *p* < 0.05, it was considered statistically significant.

### 4.9. Institutional Review Board Statement

The study protocol was approved by the Institutional Review Board of the hospital (H-1003-099-314, Seoul National University Hospital Institutional Review Board, 30 April 2010), and also conformed to the ethical guidelines of the World Medical Association Declaration of Helsinki. The biospecimens for this study were provided by the Seoul National University Hospital Human Biobank, a member of the Korea Biobank Network, which is supported by the Ministry of Health and Welfare. All samples derived from the National Biobank of Korea were obtained with informed consent under institutional review board-approved protocols.

## Figures and Tables

**Figure 1 ijms-20-01292-f001:**
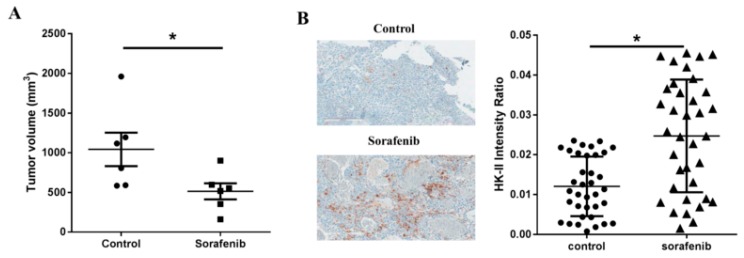
Sorafenib resulted in upregulation of hexokinase-II expression in hepatocellular carcinoma (HCC) tumors. (**A**) Murine subcutaneous HCC model of SNU-761 tumors. Sorafenib and/or 3-BP were administered two weeks after subcutaneous injection of SNU-761 cells (*n* = 5, Mann-Whitney test, * *p* = 0.043); (**B**) hexokinase (HK)-II expression was analyzed by immunohistochemical staining of specimens from each experimental group. The expression of HK-II was significantly lower in the tumors of mice receiving sorafenib + 3-BP than in the tumors of mice receiving sorafenib alone (*n* = 5, Mann-Whitney test, * *p* < 0.0001).

**Figure 2 ijms-20-01292-f002:**
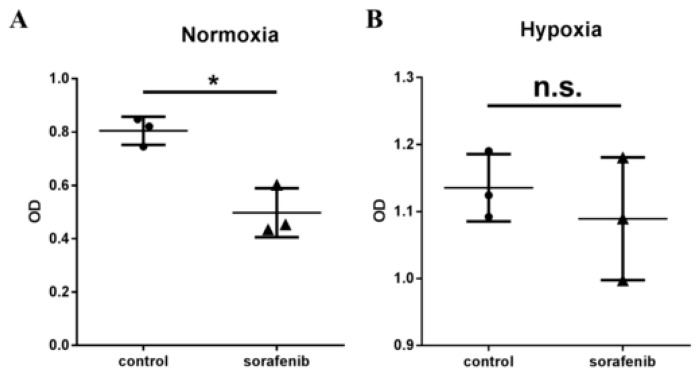
Hypoxia inhibited the effect of sorafenib on proliferation of human HCC cells.SNU-761 cells were serum starved for 16 h and then treated with sorafenib (8 μM) in the presence or absence of 3-BP (75 μM). Cell growth was determined using the MTS assay under (**A**) normoxic and (**B**) hypoxic conditions (*n* = 3, Mann-Whitney test, * *p* < 0.005 (0.0074)).

**Figure 3 ijms-20-01292-f003:**
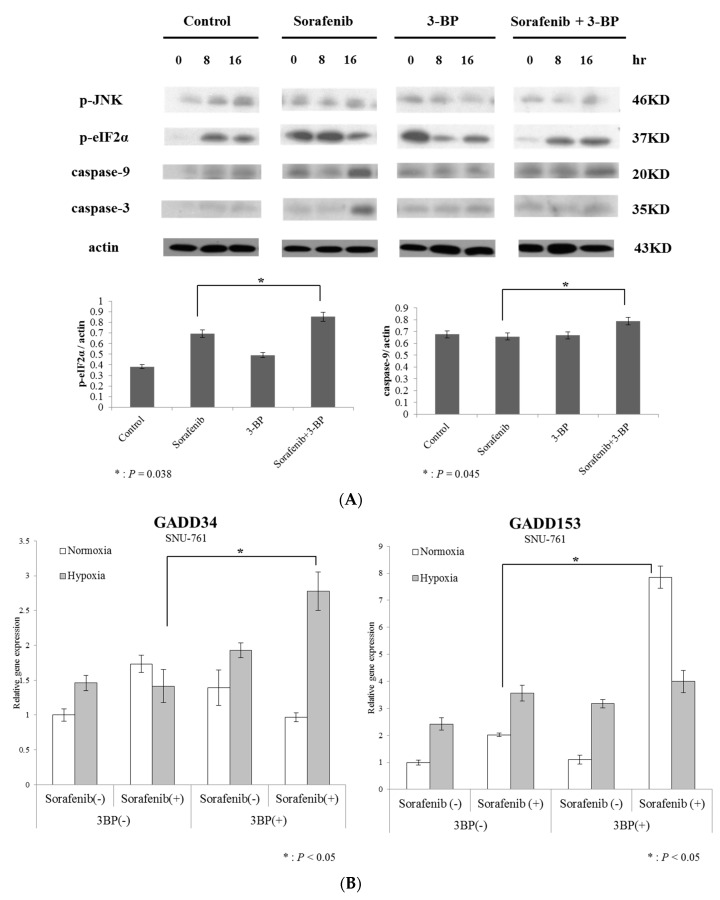
Inhibition of HK-II by 3-BP reverses increased ER stress and sorafenib resistance.The SNU-761 cells were serum starved for 16 h and then treated with sorafenib (8 μM) in the presence or absence of 3-BP (75μM). (**A**) Equivalent amounts of proteins were immunoblotted using anti-phospho-JNK, anti-phospho-eIF2α, anti-caspase-9, anti-caspase-3, and anti-actin antibodies. (**B**) The ER stress markers, *GADD34* and *GADD153*, were evaluated by quantitative real-time PCR (*n* = 3, Mann-Whitney test *t*-test, * *p* < 0.05).

**Figure 4 ijms-20-01292-f004:**
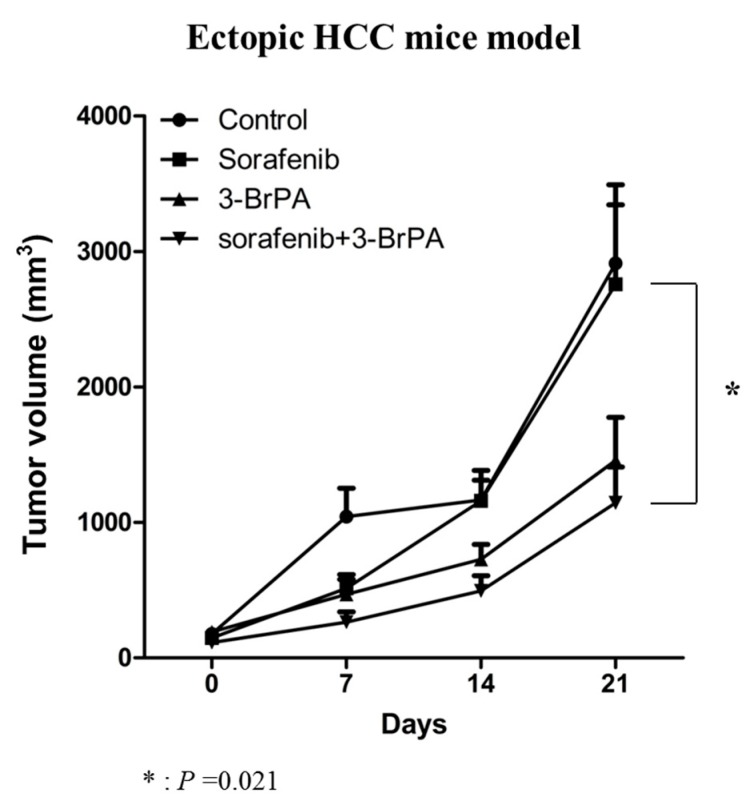
3-BP improves the anti-tumor activity of sorafenib against subcutaneous HCC tumors. Male BALB/c nu/nu mice were subcutaneously injected with 2.5 × 10^6^ SNU-761 HCC cells. The tumors in HCC xenograft mouse models were blindly measured by calipers. Two weeks after subcutaneous injection, sorafenib and or 3-BP were administered for 21 days (*n* = 8, Mann-Whitney test, * *p* = 0.032).

**Figure 5 ijms-20-01292-f005:**
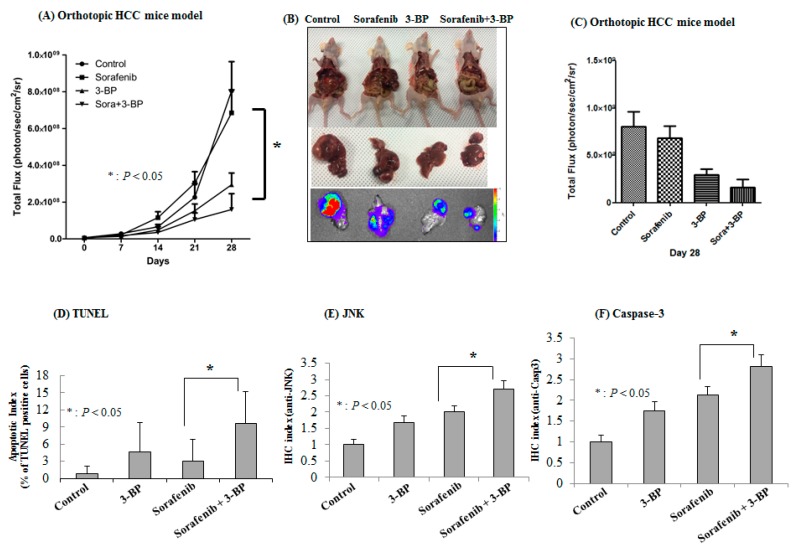
3-BP improves the anti-tumor efficacy of sorafenib against orthotopic HCC tumors. (**A**) The bioluminescent image (BLI) signals of the tumor cells from the mice receiving sorafenib, 3-BP, or sorafenib + 3-BP after 28 days of treatment. (**B**) Representative ex vivo images from autopsy. (**C**) The BLI signals evaluated immediately after autopsy. (**D**) The apoptotic index, defined as TUNEL-positive cell percentages of the total number of cells. (**E**) JNK and (**F**) capsase-3 immunohistochemical staining of sections of paraffin-embedded HCC specimens.

**Figure 6 ijms-20-01292-f006:**
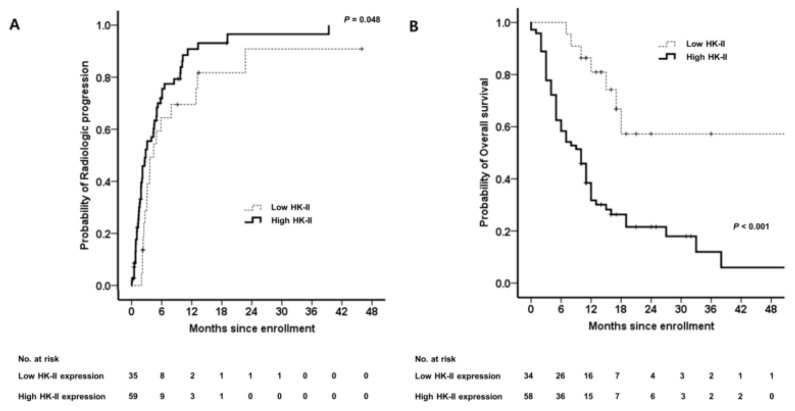
Survival analysis according to the HK-II expression level. (**A**) Time to progression. (**B**) Overall survival (OS).

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
