# Peer review of "Hexokinase-II Inhibition Synergistically Augments the Anti-tumor Efficacy of Sorafenib in Hepatocellular Carcinoma"

_ijms, 2019, doi:10.3390/ijms20061292_

Round 1
Reviewer 1 Report
Yoo et al.
IJMS-431354
Conflict of interest: none
The following report is divided into two parts containing:
A-General comments concerning the global appreciation of the work.
B-Specific comments concerning particular aspects of the article.
A-GENERAL ANALYSIS AND COMMENTS
The authors present an interesting study focusing on Hexokinase-II inhibition in the context of HCC treatment by Sorafenib, thus providing a relevant clinical context. The work relies on a range of complementary methods, assays and models from (i) in situ observations, (ii) in vitro assays, to (iii) in vivo models and (iv) human database analysis, which is well appreciated. These are comprehensively used.
However some parts lack statistical power to make the conclusions convincing and replicates must be provided to make this study more rigorous, and increase its impact. At least 6 experiments should be included in every panel to be able to perform statistics. Student tests used here with only 3 experiments can not constitute a rigorous scientific/statistic evidence.
B-SPECIFIC ANALYSIS AND COMMENTS
Major comments (5)
Major comments (1/5)
Figure 1, please use non-parametric tests since the distribution can not assumed to be normal, or alternatively assess the normal distribution with a specific test (this comment applies to other analysis).
Major comments (2/5)
Line 90 - “hypoxia, which can induce HK-II expression” please provide controls for this, it will reinforce the findings.
Major comments (3/5)
Figure 2, Please provide a description of the legend for the Y axe. Some additional details and experiments might help to interpret the results and might be provided such as (i) the density of cell from the start, (ii) kinetic experiments, and (iii) absolute cell count.
Authors could consider replicating and confirming results using Hep2G cell line commonly used as “reference” which might help the community to compare and use these results.
Major comments (4/5)
Figure 3, Please provide at least n=6 replicates to perform statistical analysis, as well as automated quantification (image J or other software).
Major comments (5/5)
Figure 5 should be properly reorganized in a one-page format. Panel A could be considered as supplement information. Please shorten the legend and avoid commenting and interpreting the results inside.
Minor comments (3)
Minor comments (1/3)
Figure 6, did the authors had the possibility to cross-analyze mutation status or histology of the tumors to understand if HKII expression might be associated with a specific tumor biology?
Minor comments (2/3)
“Supplementary Table 4. Baseline Characteristics of Study Population”
It might be relevant to provide more insight in tumor biology to present the proportion of HKII expression High / low in different stages and grades, if possible.
Minor comments (3/3)
Supplementary Table 6.
It is not very clear if all variables are included in the multiple analysis or if many have been removed by automatic selection? Also to better interpret the model, please provide the R2 and the overall significance of the model.
Author Response
Responses to the Associate Editor’s and Reviewers’ Comments
8 March, 2019
Dear reviewers and editorial staffs in International Journal of Molecular Sciences
We are sincerely grateful for your thorough consideration and scrutiny of our manuscript, “Hexokinase-II inhibition synergistically augments the anti-tumor efficacy of sorafenib in hepatocellular carcinoma”, control number ijms-455211. Through the accurate comments made by the reviewers, we better understand the critical issues in this paper. We have revised the manuscript according to the Reviewer’s suggestions. We hope that our revised manuscript will be considered and accepted for publication in the International Journal of Molecular Sciences. We acknowledge that the scientific and clinical quality of our manuscript was improved by the scrutinizing efforts of the reviewers and editors.
The changes within the revised manuscript were highlighted (underlined and in blue). Point-by-point responses to the reviewers’ comments are provided below.
Reviewer #1 :
<MAJOR COMMENTS>
1) Reviewer’s comment: Figure 1, please use non-parametric tests since the distribution cannot assumed to be normal, or alternatively assess the normal distribution with a specific test (this comment applies to other analysis).
Author’s response: We appreciate the editor’s comment. Following Reviewer’s comment, we changed the statistical method from parametric test (Student t-test) to non-parametric test (Mann-Whitney test). The Figure legends of the manuscript were now revised as follows:
“Figure 1. (A) Sorafenib and/or 3-BP were administered 2 weeks after subcutaneous injection of SNU-761 cells (n=5, Mann-Whitney test, P = 0.043) (B) The expression of HK-II was significantly lower in the tumors of mice receiving sorafenib + 3-BP than in the tumors of mice receiving sorafenib alone (n=5, Mann-Whitney test, *P < 0.0001).
Figure 2. (B) hypoxic conditions (n=3, Mann-Whitney testStudent t-test, P <0.005)
Figure 3. (B) The ER stress markers, GADD34 and GADD153, were evaluated by quantitative real-time PCR (n=3, Mann-Whitney test, P < 0.05
Figure 4. Two weeks after subcutaneous injection, sorafenib and or 3-BP were administered for 21 days (n=8, Mann-Whitney test, P = 0.032).” (page 5, lines 5–10)
2) Reviewer’s comment: Line 90 - “hypoxia, which can induce HK-II expression” please provide controls for this, it will reinforce the findings.
Author’s response: We appreciate the editor’s comment. Following Reviewer’s comment, we changed the phrase as follows;
“Since hypoxia is reported to induce HK-II expression, we investigated whether hypoxia inhibited the effect of sorafenib on the proliferation of HCC in vitro.” (page 3, lines 96–98)
3) Reviewer’s comment: Figure 2, Please provide a description of the legend for the Y axis Some additional details and experiments might help to interpret the results and might be provided such as (i) the density of cell from the start, (ii) kinetic experiments, and (iii) absolute cell count.
Authors could consider replicating and confirming results using Hep2G cell line commonly used as “reference” which might help the community to compare and use these results.
Author’s response: We appreciate the editor’s comment. Following Reviewer’s comment, we corrected the Y axis of Figure 2 as “relative growth”.
We conducted experiments on Huh7 as well as SNU761. Although HepG2, a reviewer's recommendation, is a good option, but some researchers believe that HepG2 cells are not a typical HCC cell line,[1] so we chose Huh7 instead of HepG2. We added the result of Huh-7 in Supplementary Figure S1B. The Result section of the manuscript were now revised as follows:
“We first investigated whether 3-BP combined with sorafenib inhibited proliferation of HCC cells in vitro. For this purpose, human HCC cell lines, SNU-761 (Supplementary Figure S1A) and Huh-7 (Supplementary Figure S1B) were cultured in normoxic and hypoxic state. Sorafenib inhibited cellular growth in a dose-dependent manner, and this effect of growth inhibition was even amplified when combined with 3-BP, both in normoxic and hypoxic conditions. In short, 3-BP in combination with sorafenib strongly inhibited primary tumor growth” (page 2, lines 65–70)
[1] Lopez-Terrada D, Cheung SW, Finegold MJ, Knowles BB. Hep G2 is a hepatoblastoma-derived cell line. Human pathology. 2009;40:1512-5.
4) Reviewer’s comment: Figure 3, Please provide at least n=6 replicates to perform statistical analysis, as well as automated quantification (image J or other software).
Author’s response: We appreciate the editor’s comment. Following Reviewer’s comment, we added the information of quantitative analysis using densitometry of Image J in Figure 3. A significant difference was also observed in quantitative analysis using densitometer. The statistical significance was confirmed in three experiments, so we did not increase the number of experiments.
5) Reviewer’s comment: Figure 5 should be properly reorganized in a one-page format. Panel A could be considered as supplement information. Please shorten the legend and avoid commenting and interpreting the results inside.
Author’s response: We appreciate the editor’s comment. Following Reviewer’s comment, we changed Panel A from figure to supplementary file and reorganized in one-page format. Also we shortened the legend of Figure 5. The Figure and Figure legend of the manuscript were now revised as follows:
“3-BP improves the anti-tumor efficacy of sorafenib against orthotopic HCC tumors. (A) The bioluminescent image (BLI) signals of the tumor cells from the mice receiving sorafenib, 3-BP, or sorafenib + 3-BP after 28 days of treatment. (B) Representative ex vivo images from autopsy. (C) The BLI signals evaluated immediately after autopsy. (D) The apoptotic index, defined as TUNEL-positive cell percentages of the total number of cells. (E) JNK and (F) capsase-3 immunohistochemical staining of sections of paraffin-embedded HCC specimens.” (page 8, lines 191–206)
<MINOR COMMENTS>
1) Reviewer’s comment: Figure 6, did the authors had the possibility to cross-analyze mutation status or histology of the tumors to understand if HKII expression might be associated with a specific tumor biology?
Author’s response: We appreciate the editor’s comment. We could not cross-analyze the data according to mutation status due to lacking of detailed pathology results. However, there was no significant correlation between HKII expression and tumor histology such as Edmonson grade. We added this information to the discussion section of the manuscript.
2) Reviewer’s comment: It might be relevant to provide more insight in tumor biology to present the proportion of HKII expression High / low in different stages and grades, if possible. (Supplementary table 4)
Author’s response: We appreciate the editor’s comment. We analyzed the association of HKII expression and different stage or grade, but we could not find an association. We added this information to the discussion section of the manuscript.
3) Reviewer’s comment: It is not very clear if all variables are included in the multiple analysis or if many have been removed by automatic selection? Also to better interpret the model, please provide the R2 and the overall significance of the model. (Supplementary table 6)
Author’s response: We appreciate the editor’s comment. The initial multivariable model was set using clinically relevant and significant variables in the univariate analysis (P < .05). So all variables were included in the multivariable analysis. As for the R2 and overalls significance, we added the value of R squared. The Result section of the manuscript were now revised as follows:
“High HK-II expression was independently associated with poor OS (aHR 1.882; 95% CI, 1.171–3.190; P = 0.024; R_squared by Cox & Snell 0.995) after adjustment for Child–Pugh score, tumor number, and lymph node involvement (Supplementary Table S6).” (page 9, lines 244–247)
Reviewer 2 Report
In this study, the authors examined the effect of HK-II inhibition on the efficacy of sorafenib in in vivo HCC model, and evaluated the prognostic implication of HK-II expression in patients with HCC. The authors found that upregulated HK-II expression reduced sorafenib-induced apoptotic cell death, and HK-II inhibitor (3-BP) could attenuate the reduction. These results suggest that HK-II is a promising therapeutic target for enhancing the efficacy of sorafenib and that HK-II expression might be a prognostic factor in HCC.
The findings of this study are interesting and may contribute to HCC treatment. In general, interpretation of the data is reasonable; however, there are still several issues need to be addressed.
Comments:
1. The majority of the experiments were done in one HCC cell line (SNU-761). The author should perform the experiments with more cell lines to provide stronger evidence.
2. Figure 3B: More ER-stress markers should be examined.
3. Scale bar in all microscopy should be added.
4. Figure 3 and 5: The labels of some subgraphs are missing.
5. The manuscript contains many grammatical and syntax errors and must be carefully edited.
Author Response
Responses to the Associate Editor’s and Reviewers’ Comments
8 March, 2019
Dear reviewers and editorial staffs in International Journal of Molecular Sciences
We are sincerely grateful for your thorough consideration and scrutiny of our manuscript, “Hexokinase-II inhibition synergistically augments the anti-tumor efficacy of sorafenib in hepatocellular carcinoma”, control number ijms-455211. Through the accurate comments made by the reviewers, we better understand the critical issues in this paper. We have revised the manuscript according to the Reviewer’s suggestions. We hope that our revised manuscript will be considered and accepted for publication in the International Journal of Molecular Sciences. We acknowledge that the scientific and clinical quality of our manuscript was improved by the scrutinizing efforts of the reviewers and editors.
The changes within the revised manuscript were highlighted (underlined and in blue). Point-by-point responses to the reviewers’ comments are provided below.
Reviewer #2:
<GENERAL COMMENTS>
1) Reviewer’s comment: The majority of the experiments were done in one HCC cell line (SNU-761). The author should perform the experiments with more cell lines to provide stronger evidence.
Author’s response: We appreciate the editor’s comment. Following Reviewer’s comment, we added the result of Huh-7 in Supplementary Figure S1B. The Result section of the manuscript were now revised as follows:
“We first investigated whether 3-BP combined with sorafenib inhibited proliferation of HCC cells in vitro. For this purpose, human HCC cell lines, SNU-761 (Supplementary Figure S1A) and Huh-7 (Supplementary Figure S1B) were cultured in normoxic and hypoxic state. Sorafenib inhibited cellular growth in a dose-dependent manner, and this effect of growth inhibition was even amplified when combined with 3-BP, both in normoxic and hypoxic conditions. In short, 3-BP in combination with sorafenib strongly inhibited primary tumor growth” (page 2, lines 65–70)
2) Reviewer’s comment: Figure 3B: More ER-stress markers should be examined.
Author’s response: We appreciate the editor’s comment. We checked ER stress markers such as JNK and p-eIF in Figure 3A and further validated GADD34 and GADD 153 in Figure 3B. For this reason, we carefully think that we already tested variable ER-stress markers in Figure 3.
3) Reviewer’s comment: Scale bar in all microscopy should be added.
Author’s response: We appreciate the editor’s comment. Following Reviewer’s comment, we added scale bar in all microscopy.
4) Reviewer’s comment: Figure 3 and 5: The labels of some subgraphs are missing.
Author’s response: We appreciate the editor’s comment. Following Reviewer’s comment, we added labels of subgraph. The Y-axis label of Figure 3B was added as “relative gene expression”. We added the labels of Figure 5, and reorganized in one-page format.
5) Reviewer’s comment: The manuscript contains many grammatical and syntax errors and must be carefully edited.
Author’s response: We appreciate the editor’s comment. Following Reviewer’s comment, we conducted the English proofreading and corrected grammatical and syntax errors as much as possible. We have attached certification file of English editing by MDPI.
Round 2
Reviewer 1 Report
The authors significantly implemented changes to improve the quality of the manuscript according to comments,
Reviewer 2 Report
The authors have addressed all questions raised during the first reviewing procedure. I have found the manuscript to be much improved and recommended acceptance.